# Perceived Impact of Taiwan’s National Health Insurance Allocation Strategy: Health Professionals’ Perspective

**DOI:** 10.3390/ijerph16030467

**Published:** 2019-02-05

**Authors:** Patrick Opiyo Owili, Miriam Adoyo Muga, Ya-Ting Yang, Yi-Hsin Elsa Hsu

**Affiliations:** 1Department of Public Health, School of Health Sciences, University of Eastern Africa, Baraton, 30100 Eldoret, Kenya; owilip@ueab.ac.ke; 2School of Health Care Administration, Taipei Medical University, Taipei 110, Taiwan; 3Department of Foods, Nutrition and Dietetics, School of Science and Technology, University of Eastern Africa, Baraton, 30100 Eldoret, Kenya; mugam@ueab.ac.ke; 4Center of General Education, Taipei Medical University, Taipei 110, Taiwan; 5Biotechnology Executive Master‘s Degree in Business Administration (BioTech EMBA), Taipei Medical University, Taipei 110, Taiwan

**Keywords:** health insurance, resource allocation, global budget, health professionals, perceptions and attitudes, structural equation modeling

## Abstract

Studies on health care demand have indicated high levels of public satisfaction with Taiwan’s National Health Insurance (NHI). However, the global budget allocation mechanism (GBAM) used by NHI has led to various adjustments in the providers’ way of practice, quality of care, utilization of care, and health expenditure. Studies focusing on the satisfaction of providers with health care supply, however, remain limited. We therefore explored the provider’s perceived impact of the NHI allocation plan. A cross-sectional data of 299 health professionals was collected at Taipei Medical University Hospitals in April 2012. Perceptions and attitudes were assessed using a validated 5-point Likert-type questionnaire before using a structural equation modeling technique to explore the complex interrelationships of the NHI’s perceived impact. The causal path relationships between the latent variables ‘characteristics of NHI’s allocation plan’ and ‘perceived positive effect’ (*β* = 0.39), ‘perceived positive effect’ and ‘satisfaction of health professionals’ (*β* = 0.53), and between ‘characteristics of NHI’s allocation plan’ and ‘satisfaction of health professionals’ (*β* = 0.30) were positively associated; while the path relationships between the latent variables ‘perceived negative effect’ and ‘satisfaction of health professionals’ (*β* = −0.27) and ‘characteristics of NHI’s allocation plan’ and ‘attitude toward allocation criteria’ (*β* = −0.22) were negatively associated. These results indicate that providers perceived a positive impact of the NHI allocation strategy. The NHI allocation plan is an important decision-making tool among policy makers since it helps optimize outcomes. Research based on its impact at both horizontal and vertical levels on the supply side may be useful towards understanding Taiwan’s GBAM. Policy-makers should therefore consider understanding the impact of GBAM at both the demand and supply side in adjusting allocation criteria.

## 1. Introduction

Four decades have passed since Saward [1] predicted that National Health Insurance (NHI) would lead to a more regulated resource allocation criterion which would then force a reduction in the physician ratio. Today, the ratio of health professionals has since increased over time in the health systems operating under the NHI (i.e., Britain) only to be faced with new challenges such as limited health care resources and, hence, rationalization of health care standards [2]. Equity and scarcity in health care resources have remained some of the most important concerns to both policy makers and health professionals who have reported unequal distribution of resources among different sectors and groups [3,4,5]. Consequently, health care providers desire to be involved in the allocation and rationalization of these scarce health care resources [6]. Since these professionals occupy a position which allows them to notice the impact of policies that affect their clinical practice, it would be important if their input is regarded when pooling and allocating the scarce health care resources. Health professionals may change their mode practice depending on the health care financing and payment mechanisms available.

In Taiwan, for example, health care providers increased their service volume as a result of the single payer system and the NHI’s global budget allocation mechanism (GBAM), which subsequently led to an increase in the utilization of care and a surge in the overall health expenditure [7]. The NHI has since moved along the spectrum of second-generation insurance plans from the year 2013 so as to increase its revenue base by levying a supplementary premium on the income from professional practice, high bonuses, stock dividends, interests, rentals, and part-time jobs. Nevertheless, the NHI’s allocation criteria still remained the same. It is, however, not clear how the health care providers responded to these changes that may as well lead to an increase in health care spending over time [8]. This study, therefore, explored the perceived impact of GBAM from the perspective of health providers in Taiwan.

Taiwan is a high-income country with a multi-party democracy since the year 1993. The country had its health care challenges that later sparked the changes in its health care system in mid 1990s. Some of its current challenges include an aging society and non-communicable diseases (NCDs) similar to other high-income countries. The nation’s health care financing relies on a single payer system designed to be financially sufficient and responsible for its deficits from the premium collected which is directly proportional to individual’s income bracket. Since the NHI’s inception in March 1995, 99% of its 23 million populace enjoys an almost free access to health care services with only a small co-payment required by clinics and hospitals, of which over 60% are privately operated [7,9]. With the current escalating health expenditure, it may be difficult to attain an efficient and equitable allocation strategy that may cater for all the health care services provided. Hence, health professionals may not execute their duties effectively if resources are constrained. 

The strategy used in pooling and allocating health care resources may also remain a major challenge not only to the policy makers but also to the health professionals who are at the forefront in the health system [10,11,12]. Some policies may either limit these professionals from acting as a perfect agent to their patients or may sometimes encourage innovation in health care. In an attempt to fill the gap on the perceived impact of GBAM, our study explored the provider’s perceived impact of the NHI’s allocation criteria. This would give policy makers an insight towards improving the allocation mechanism, which would then consider both the health care demand side and supply side in the health system. This is the first study, to the best of our knowledge, that has examined the relationship between the NHI allocation plan and health professionals’ perception, satisfaction, and attitude in a high-income country. The only documented study is in a low-income country [13].

### Taiwan NHI’s Risk Pooling and Allocation Strategy

Taiwan created a health financing system suitable for its population after evaluating the strengths and weaknesses of different health systems around the world. Today, its health care system is one of the best globally [14], which has been described elsewhere [15]. Taiwan’s NHI is a government-run social insurance which is responsible for collecting premium and allocating and purchasing health care services for its general population. In 2008 and 2014, the NHI budget accounted for 3.1% and 6.6%, respectively, of the total government budget, thereby indicating a 100% increase in its health budget within a 6 year period. This was the result of an increased utilization of health care services. Even though the NHI faced a considerable resistance at inception as seen in the 40% public satisfaction rate, the rates have since then increased to over 80% in recent years. 

The main source of NHI funds is income-based premium from employees (payroll taxes), employers, and government contribution. The premium rate charged on an individual’s income increased from 4.25% (at inception) to 4.55% and 5.17% in 2002 and 2010, respectively, but was later reduced to 4.91% in 2013 [16]. For employees in public or private enterprises, a share of premium is usually paid by the employee himself, the employer, and the government at the rates of 30%, 60%, and 10%, respectively. On the other hand, the self-employed with high-income is required to pay 100% of the premium. The government pays 40% of the premium for the unemployed and 100% of the premium for military personnel and low-income households. The premium amount also varies according to the number of dependents, of which those in excess of three dependents are insured for free. The collected premium is also supplemented with the contributions from co-payments and taxes levied on alcohol, tobacco, lottery, environmental pollution, and automobile insurance funds [17,18]. Once the premium is collected, the NHI allocates and distributes resources using GBAM to health care providers. 

Before 1998, the NHI used a fee-for-service to pay providers, but in an effort to contain the escalating health expenditure, it introduced the GBAM—a payment mechanism in which prices are determined *ex post* on a point value—and then providers are paid according to the budget to cover for the operating costs [19]. The global budget (GB) was introduced for dental services, Chinese medicine, Western medicine in community clinics, and Western medicine in hospital services (i.e., outpatient and inpatient services) in 1998, 2000, 2001, and 2002, respectively. In allocating health care resources, the NHI’s Medical Expenditure Negotiation Committee (MENC) negotiates the expenditure cap prior to the next fiscal year, and a ‘floating point value’ (FPV) on the predetermined budget is then set based on various indicators as represented in the following equation [20]:
(1)$${ℰ}_{𝓉}=\sum_{𝒾=1}^{I}\left[\beta_{𝒾}U_{𝓉𝒾}\right]\times A$$
where ${ℰ}_{𝓉}$ is the total NHI budget, negotiated and predetermined based on the previous year’s expenditure; $\beta_{𝒾}$ is the variable service volume,’$U_{𝓉𝒾}$ is unit prices of all service items listed on the NHI payment scheme; and A is the FPV to adjust for the total claimed expenditure in order to meet the predetermined national budgeted expenditure before reimbursement of the providers is done. Chen and Chuang [21] have provided the details of recent medical spending.

The providers would be required to file their monthly claims first. Then, the total claimed expenditure would be calculated based on the sum of the service volume per item multiplied by the unit price of each of the listed service items. It is then adjusted using the FPV that is always determined every three months after a review of all hospitals’ claims. The FPV is negatively related to the countrywide service volume (patient volume and medical orders); that is, the higher the service volume, the lower the FPV for all hospitals in that year.

However, with the improved access to health care services as a result of the reduced financial barriers, providers changed their ways of operation so as to adapt to the new system. This led to an increase in service volume and hospitalization, patient’s length of stay, number of prescribed procedures, claimed expenses, co-payment fee per admission, and the degree of acceptability in small and large hospitals [7,9,20,22]. The studies on the providers’ behavior, though plausible, only explored changes in their way of operation but did not explore their satisfaction, attitude, and what they perceived to be the impact of the NHI’s allocation plan. Nevertheless, several other studies have been performed to explore the attitudes and perceptions of health professional on other issues such as use of analgesics, sexual dysfunction, safety culture, use of complementary medicine, and nutrition [23,24,25,26,27] but not on the NHI’s allocation strategy. In this study, therefore, we explored and provided a useful insight regarding health professionals’ perceived impact of GBAM, and we also validated the structural model explaining these causal relationship [13].

## 2. Methods

### 2.1. Study Site and Sample Size

This cross-sectional study was undertaken at Taipei Medical University (TMU) hospital, a private medical teaching hospital with three main hospitals (the on-campus hospital, the TMU Wan Fang Medical Center, and TMU Shuang Ho hospital) that form a medical care ‘golden triangle’ within Taipei city. The hospitals have a total capacity of 3000 beds, with the on-campus hospital and Wan Fang Medical Center being 5 kilometers apart and always working together in clinical teaching, research, human resource, and medical services. We restricted data collection to a sample frame of 2264 employees in the on-campus hospital to avoid double collection of data from health professionals (i.e., physicians, nurses, pharmacists, nutritionists, laboratory technologists, and others) who serve in all three hospitals. We used an online sample size calculator to obtain the minimum sample size of 329 by adapting a 5% error of margin, a 95% confidence interval, and a 50% response distribution [28]. A total of 318 questionnaires were successfully returned, of which 19 of them were not fully completed and were therefore excluded from our analyses. Hence, a 90.9% response rate was realized. This sample size was considered adequate for our final analyses because the minimum satisfactory sample size for performing a structural equation modeling (SEM) procedure is usually between 100 and 150 [29,30].

### 2.2. Structural Model and Instrument

To examine the relationship between NHI’s allocation strategy and its perceived impact, we used similar model published elsewhere [13]. The structural model has five latent variables related in different ways and include concepts of the allocation strategy, perceived effect (positive and negative), health professionals’ satisfaction, and attitude on the allocation criteria. A validated self-administered 5-point Likert-type questionnaire was employed to assess health professionals’ perceptions and attitudes on NHI’s allocation criteria [13]. However, the questionnaire items were translated into Traditional Chinese (the national and official language in Taiwan) by a team of three experts. Some of the questionnaire items were adjusted to reflect the context of Taiwan’s health system. Medical students were then recruited and trained to assist in data collection before collecting data in the month of April 2012. 

### 2.3. Measures

The four dependent latent variables adopted from the previous study were measured using the following indicators: (i) a perceived positive effect (PPIAM) latent variable that had four measurements including improved wages and living standards (Pec2_16), human resource increase (Pef2_15), modern equipment acquisition (Pde2_3), and distribution of resources equally (Peq2_13b); (ii) a perceived negative effect (PNIAM) latent variable was measured using unachievable universal coverage (Nec2_12d), increased hospital length-of-stay (Nef2_6), scanty amenities (Nde2_7), and increased cost of care (Neq2_12e); (iii) a satisfaction of health professionals (OPS) latent variable had six indicators including whether the professional is satisfied with the wages and living standards (Sec3_3d), performance of hospital (Sef3_3c), available services (Shc3_1b), costs of service (Seq3_3a), information available (Sjo3_2c), and resources available (Sjo3_2a); and finally (iv) an attitude toward allocation criteria (ARAM) latent variable included what the allocation strategy should consider such as hospital size (Aal4_10d), cost of services (Aeq4_9e), human resource (Aec4_11a), equipment and facilities (Ade4_11b), and patient volume (Ah4_9b) [13]. The codes (in parenthesis) were used in the generation of the path diagram.

The main independent latent variable, the characteristics of the NHI’s allocation plan (BNHCRAM), included important features of a good resource allocation criterion [31]. The important characteristics identified if the allocative strategy improves overall performance (Bpe4_7), supports economic and health sector development (Bde4_6), cater for unmet needs (Bun4_5), improves equity (Beq4_4), improves allocative efficiency by accounting for health care variations (Bef4_3), meet the health care needs and demand (Bhc4_2), and is less complex (Bcx4_1). The codes (in parenthesis) were also adopted to generate the path diagram using Linear Structural Relationship (LISREL) software (Scientific Software International Inc., Skokie, IL, USA).

### 2.4. Analysis

We used a structural equation modeling (SEM) technique in LISREL software version 8.72 (Scientific Software International, Inc., Lincolnwood, IL, USA), a tool used for analyzing complex relationships [32]. Data were first screened for univariate and multivariate normality before performing a reliability and consistency analysis in SPSS version 20 (IBM Corp., Armonk, NY, USA). We adopted a Cronbach’s alpha of 0.70 and above as the acceptable level of reliability and consistency [33]. The alpha level for the five latent constructs were very high and are as follows: perceived positive effect (0.80); perceived negative effect (0.80); satisfaction of health professionals (0.81); attitude toward allocation criteria (0.81); and characteristics of NHI’s allocation plan (0.87).

We then employed a confirmatory factor analysis (CFA) to check for the dimensionality of the measures and to confirm the a priori model [13]. We used a Robust Maximum Likelihood (RML) estimation procedure, which has the assumption of multivariate non-normality, to estimate asymptotic covariance matrix of the structural model (see Appendix A for the covariance matrix of the latent variables) [34]. The model’s regression weights are presented as parameter estimates that estimated the effect of measurement and latent variables. The fit indices and the parameter estimates were finally assessed and evaluated for statistical significance, magnitude of the effect (including the direction), and causality [35].

### 2.5. Ethical Statement

The ethical approval to conduct the study was obtained from the Joint Independent Review Board (JIRB) of TMU Hospital (approval no. 201012013) before data collection. Participation in this study was voluntary and the participants were required to sign an informed consent that was attached to the questionnaire as an indication of willingness to participate. The respondents were also informed that they had a right to withdraw from the study if they opted not to continue. All data were anonymized.

## 3. Results

### 3.1. Descriptive Statistics

Table 1 presents the characteristics of the survey respondents. The majority of the respondents were female (68.6%), below 30 years of age (57.5%), and have worked for less than 6 years (53.8%). The nurses were 47% of the overall respondents while the physicians accounted for 27.8%. Moreover, the majority of the participants were not religious, with less than half of the believers being Christians. 

### 3.2. Structural Equation Modeling

In the model evaluation, the incremental fit indices were high and leaning towards a perfect fit compared to the absolute fit indices (see Appendix A). The χ^2^ test (χ^2^ = 2495, *df* = 290) was significant at *p* < 0.001; but, since the χ^2^ test is often sensitive to sample size, the root-mean-square error of approximation (RMSEA) was subsequently considered. Traditionally, a RMSEA below 0.08 indicates a good model fit [36]. Hence, the RMSEA of 0.08 in this study showed that the data fit the model. We therefore proceeded to the evaluation of the measurements and the structural model.

All parameter estimates of measurement variables were statistically significant at *p* < 0.001, except for the variable ‘hospital length-of-stay increased’ in the latent construct ‘perceived negative effect’ (Table 2). Figure 1 also presents standardized estimates of the structural model including the causal relationships between the latent constructs.

In the evaluation of the causal path relationship, Figure 2 presents significant relationships, with the effect-size and direction of the estimate also summarized in Table 3. Five path relationships were statistically supported while four were rejected. Path relationship 1 (i.e., from ‘characteristics of NHI’s allocation plan’ to ‘perceived positive effect’) was positive (*β* = 0.39) and significant at *t* = 4.83 (*p* < 0.001), while path relationship 2 (i.e., from ‘characteristics of NHI’s allocation plan’ to ‘perceived negative effect’) was rejected at *t* = −0.14 (Table 3).

Path relationships 3 (i.e., from ‘perceived positive effect’ to ‘satisfaction of health professionals’) and 5 (i.e., from ‘characteristics of NHI’s allocation plan’ to ‘satisfaction of health professionals’) were also statistically significant and positively related, with *β* = 0.53 (*t* = 7.00, *p* < 0.001) and *β* = 0.30 (*t* = 6.00, *p* < 0.001), respectively. Path relationships 4 (i.e., from ‘perceived negative effect’ to ‘satisfaction of health professionals’) and 9 (i.e., from ‘characteristics of NHI’s allocation plan’ to ‘attitude toward allocation criteria’) were statistically supported and related negatively at *β* = −0.27 (*t* = −4.05, *p* < 0.001) and *β* = −0.22 (*t* = −2.31, *p* = 0.02), respectively.

However, path relationships 6 (i.e., from ‘perceived positive effect’ to ‘attitude toward allocation criteria’), 7 (i.e., from ‘satisfaction of health professionals’ to ‘attitude toward allocation criteria’), and 8 (i.e., from ‘perceived negative effect’ to ‘attitude toward allocation criteria’) were not statistically significant at *p* < 0.05.

## 4. Discussion

Our study aim was to explore the perceived impact of NHI’s GB allocation criteria among health professionals and to test a structural model that had examined the relationship between characteristics of the allocation criteria, perceived positive and negative effect, health professionals’ satisfaction, and attitude on the allocation criteria. The results in this study supported five path causal relationships that were also found to be significant in the previous study while four were not supported at *p* < 0.05 [13].

The study finding supported a causal relationship between the latent constructs ‘characteristics of NHI’s allocation plan’ and ‘perceived positive effect.’ This finding aligns with common sense that health professionals who identify positive characteristics of the health care allocation criteria would be more optimistic of its impact thereof and vice versa. Even though the previous study supported the association between the latent constructs ‘characteristics of NHI’s allocation plan’ and ‘perceived negative effect’ at *t* = 9.18 (*p* < 0.001) and *β* = 1.14 [13], this study did not find a significant association between the two latent constructs. It could be that the relationship between ‘characteristics of NHI’s allocation plan’ and ‘perceived negative effect’ was strongly dependent on the context and level of health system development. Health professionals in this study viewed the GB allocation strategy as a positive contributing factor to the equity, efficiency, development, and the economy of Taiwan as indicated by the positive association between ‘characteristics of NHI’s allocation plan’ and ‘perceived positive effect.’. Since Taiwan’s NHI reduced the financial barriers of health care access, the providers’ practice also changed, and this subsequently led to an increase in health care utilization and health expenditure as well as providers’ income [7,9,20,22]. Hence, the income effect may have partly explained the positive responses of health professionals. However, the role of the NHI on moral hazard still needs to be evaluated. 

Path relationship 3 (i.e., from ‘perceived positive effect’ to ‘satisfaction of health professionals’) was supported with a significance level of *t* = 7.00 (*p* < 0.001) and *β* = 0.53 (Table 3). This finding suggests that the positive effect of the allocation criteria contributed to the satisfaction of health professionals in Taiwan. The positive effect of NHI’s resource allocation criteria is generally known as a significant aspect towards improving both the population health and health care strategy. It converges with the intuition that if the professionals perceive that the allocative strategy has some negative impact, then it should be negatively related with satisfaction, i.e., path relationship 4, which was also statistically significant at *t* = −4.05 (*p* < 0.001) and *β* = −0.27. Even though professionals’ satisfaction may not be a valuable concept for evaluating effects of the perceived impact of the allocation criteria, these relationships were supported in both studies [13]. However, the reason for this significant relationship may be due to the workload effect experienced by health professionals in Taiwan. For example, one respondent noted that “the hospitals are competing for NHI reimbursement, and hence they require health professionals to do more by serving as many patients as possible.” The workload effect may also lead to dissatisfaction among the professionals. The implications of this finding suggest that both perceived positive and negative effects are relevant to the understanding of the satisfaction of health professionals but may be irrelevant to the understanding of the function of GBAM [37]. Since Taiwan’s NHI has an outstanding transformation history and development [38], future research efforts need to consider different items regarding its impact at different times and levels. In other words, the length of time and history of the NHI existence and development may be relevant towards understanding health professionals’ perceptions, satisfaction, and attitudes. 

Path relationship 5 (i.e., from ‘characteristics of NHI’s allocation plan’ to ‘satisfaction of health professionals’) was also supported with an optimum level of significance at *t* = 6.00 (*p* < 0.001) and *β* = 0.30. Meanwhile, path relationship 9 (i.e., from ‘characteristics of NHI’s allocation plan’ to ‘attitude toward allocation criteria’) was statistically supported at *t* = −2.31 (*p* = 0.02) and *β* = −0.22. Today, Taiwan’s NHI is faced with an escalating pressure due to the fast aging populace and other economic-related challenges that led to the introduction of supplementary premium on the capital gains, and this indicates the unfinished NHI’s reform agenda. A new reform in its allocation strategy that may further lead to change in a providers’ way of operation may require involving health professionals at all levels who have experienced the effect of GBAM. Even though this study did not support path relationships 6 and 7, which were statistically significant in the previous study, path relationship 8 was, however, rejected in both studies [13].

Our study had limitations and strengths. Although some vital relationships were sustained in this study, the limitations include the inability to generalize the findings because it focused not only on one health care entity but also on a sample that may have not been representative of the views of all health professionals (e.g., over 50% of respondents were aged 30 years and below and had worked for less than 6 years). These professionals who almost all serve in the three hospitals may as well give a biased opinion since their level of experience as a result of rotation may be different. The cross-sectional nature of the study may also have reflected the momentary experiences of these professionals unlike a cohort study that may consider variations over time. Two strengths are however noted. First, this is the first study, to the best of our knowledge, to examine the relationship between GB allocation strategy and its perceived impacts, satisfaction, and attitude among health professionals in Taiwan. Second, the analytical strategy that we used gives it strength because it allowed us to examine the complex relationships in the intricate model. Previous research also found that the health care resource allocation strategy is directly related to the degree of providers’ satisfaction and attitude, which partially explains the findings in this study. Future studies should therefore address the various impacts of the allocation strategy on clinical practice and population health at both horizontal and vertical levels using either a similar or a different method that will extend the findings of this study.

## 5. Conclusions

In conclusion, we found that ‘characteristics of NHI’s allocation plan’, ‘perceived positive effect’, and ‘perceived negative effect’ were associated with ‘satisfaction of health professionals’ which in turn was not directly related with ‘attitude toward allocation criteria’. However, only the latent construct ‘characteristics of NHI’s allocation plan’ was directly associated with ‘attitude toward allocation criteria.’ The associations between these latent variables were like the findings of the previous study except that the direction of the parameters were different [13]. Moreover, in this study, the health care resource allocation criterion that is viewed as a decision-making tool can help optimize outcomes but may not be seen as the actual process for improving health outcomes. These findings suggest that health professionals in Taiwan agree that the NHI’s GB allocation strategy has a positive impact on equity, development, efficiency, and on the economy, echoing the over-80% public satisfaction rate found in the national survey [16]. The negative relationship between the characteristics of the NHI’s allocation plan and attitude towards what the allocation strategy should consider indicate that GB restricts patient volume, hospital size, human resource, and available facilities and equipment.

As the Taiwanese population remains satisfied with the NHI at rates over 80%, the challenge associated with quality of care while controlling the escalating national health expenditure may be the next issue. Health professionals may be working long hours as a result of the increased utilization of heath care services, and yet quality of care may also be compromised. Policy makers should therefore consider evaluating not only the satisfaction on the demand side of health care but also on the supply side both vertically and horizontally so as to extend the findings of this study. An optimal point of improving the quality of care at a minimal cost is important in maintaining the population’s general health outcome. Consequently, the efficiency of the Taiwanese health care system, in relation to quality of care, should be an area of focus in future studies.

## Figures and Tables

**Figure 1 ijerph-16-00467-f001:**
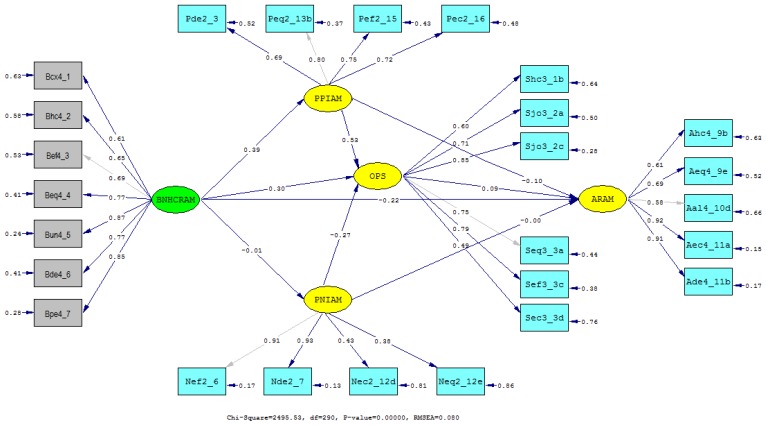
Standardized estimates of the structural equation model. The measurement indicators are in a rectangular shape while the latent constructs are in an oval shape. The arrow from one latent construct to another indicates the direction of the relationship.

**Figure 2 ijerph-16-00467-f002:**
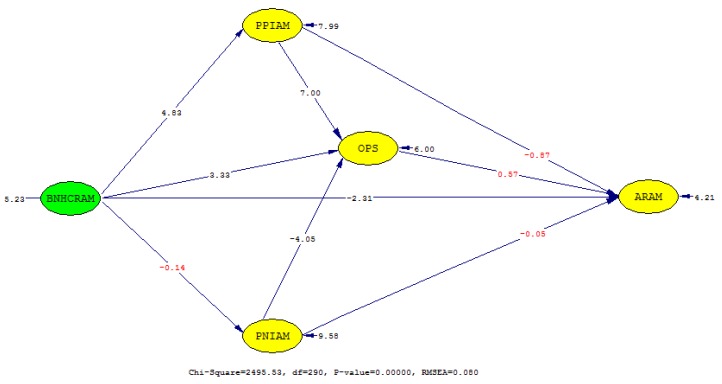
The *t*-values of the relationships in the structural model. The values appearing in red are non-significant. The latent constructs are in an oval shape while the arrow from one latent construct to another indicates the direction of the relationship.

**Table 1 ijerph-16-00467-t001:** Characteristics of health professionals who responded to the survey.

Variables	(n = 299)	%
Gender		
Male	94	31.4
Female	205	68.6
Age groups (years)		
≤30	172	57.5
31–40	100	33.4
41–50	22	7.4
≥51	5	1.7
Years of experience		
≤5	161	53.8
6–10	69	23.7
11–15	47	15.7
16–20	17	5.7
≥21	5	1.7
Specialty		
Physician	83	27.8
Nurses	141	47.2
Pharmacist	27	9.0
Nutritionist	5	1.7
Other	43	14.4
Religious status		
Non-believer	168	56.2
Believer	131	43.8
Religion (believers only)		
Christianity	60	45.8
Buddhism	50	38.2
Other	21	16.0

**Table 2 ijerph-16-00467-t002:** Standardized parameter estimates of measurement indicators by latent construct.

Codes	Indicator	Standardized Estimate	Standard Error	*t*-Value (n = 300)
**PPIAM**	**Perceived positive effect**			
Pec2_16	Improved wages and living standards	0.72	0.08	11.17 ***
Pef2_15	Human resource increase	0.75	0.05	14.40 ***
Peq2_13b	Distribution of resources equally	0.80	0.24	4.25 ***
Pde2_3	Modern equipment acquisition	0.69	0.06	11.69 ***
**PNIAM**	**Perceived negative effect**			
Nec2_12d	Unachievable universal coverage	0.43	0.07	7.02 ***
Nef2_6	Hospital length of stay increased	0.91	0.21	1.61
Nde2_7	Scanty amenities	0.93	0.10	10.94 ***
Neq2_12e	Cost of care increased	0.38	0.07	6.12 ***
**OPS**	**Satisfaction of health professionals**			
Sec3_3d	Wages and living standards	0.49	0.09	6.81 ***
Sef3_3c	Performance of hospital	0.79	0.06	18.83 ***
Shc3_1b	Available services	0.60	0.09	8.71 ***
Seq3_3a	Service costs	0.75	0.14	5.69 ***
Sjo3_2c	Available information	0.85	0.05	15.97 ***
Sjo3_2a	Available resources	0.71	0.06	12.45 ***
**ARAM**	**Attitude toward allocation criteria**			
Aal4_10d	Hospital size consideration	0.58	0.02	6.85 ***
Aeq4_9e	Cost of services consideration	0.69	0.13	9.02 ***
Aec4_11a	Human resource consideration	0.92	0.39	8.05 ***
Ade4_11b	Consider equipment and facilities	0.91	0.27	17.23 ***
Ahc4_9b	Patient volume consideration	0.61	0.18	7.42 ***
**BNHCRAM**	**Characteristics of NHI’s allocation plan**			
Bpe4_7	Improves overall performance	0.85	0.23	11.35 ***
Bde4_6	Supports economy and heath sector development	0.77	0.23	11.35 ***
Bun4_5	Cater for unmet needs	0.87	0.16	9.06 ***
Beq4_4	Improves equity	0.77	0.09	14.09 ***
Bef4_3	Improves allocative efficiency	0.69	0.06	4.97 ***
Bhc4_2	Meet health care needs and demand	0.65	0.11	9.32 ***
Bcx4_1	Less complex	0.61	0.22	6.65 ***

NHI, National Health Insurance; *** *p* < 0.001 (two-tailed).

**Table 3 ijerph-16-00467-t003:** Parameter estimates and direction of the relationships in the structural model.

No.	Path	Standardized Estimate	Standard Error	*t*-Value
1	BNHCRAM → PPIAM	0.39	0.20	4.83 ***
2	BNHCRAM → PNIAM	−0.01	0.20	−0.14
3	PPIAM → OPS	0.53	0.06	7.00 ***
4	PNIAM → OPS	−0.27	0.05	−4.05 ***
5	BNHCRAM → OPS	0.30	0.07	6.00 **
6	PPIAM → ARAM	−0.10	0.02	−0.87
7	OPS → ARAM	0.09	0.04	0.57
8	PNIAM → ARAM	−0.001	0.01	−0.05
9	BNHCRAM → ARAM	−0.22	0.05	−2.31 *

* *p* < 0.05; ** *p* < 0.01; *** *p* < 0.001 (two-tailed).

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
