# Peer review of "Perceived Impact of Taiwan’s National Health Insurance Allocation Strategy: Health Professionals’ Perspective"

_ijerph, 2019, doi:10.3390/ijerph16030467_

Round 1

Reviewer 1 Report

This is an interesting paper with complex modelling of economic resource data.

There are a few comments:

1). Many sentences are too long and complex, these need to be adjusted for readability.

2). There are many instances where the English is not correct and should be corrected.

3). Use either English spelling or American English but not both (e.g. some sections use a 'z' such as rationalize (rationalise) and then the word ageing (aging) is used.

4). I do not feel able to comment on the complexity of the economic modelling as this is not my area, however the ability to investigate resource use and benefit is useful.

5). I am not sure the data on religious beliefs adds anything to the paper.

Author Response

Dear reviewer,

Thanks for your valuable comments and the time you took to review our manuscript. We have carefully considered your comments and incorporated them into our revision (See also the revisions in tracked changes). We also note that the manuscript has been checked for grammatical mistakes, spelling, punctuation and improvement of style using a professional English editing services.

We have replied your comments by point as below. Please note the following:

-          Our responses herein follow >>>

Yours sincerely,

Yi-Hsin Elsa Hsu

================

Comments and Suggestions for Authors

This is an interesting paper with complex modelling of economic resource data.

>>>Thank you for the comment.

There are a few comments:

1). Many sentences are too long and complex, these need to be adjusted for readability.

>>>Thank you. The authors have addressed the reviewers concern and the manuscript has been adjusted accordingly using professional English editing services.

2). There are many instances where the English is not correct and should be corrected.

>>>Thank you. The document has been corrected accordingly using professional English editing services.

3). Use either English spelling or American English but not both (e.g. some sections use a 'z' such as rationalize (rationalise) and then the word ageing (aging) is used.

>>>Thank you for the important suggestion. We have corrected the manuscript and used American English in the entire document.

4). I do not feel able to comment on the complexity of the economic modelling as this is not my area, however the ability to investigate resource use and benefit is useful.

>>>Thank you. Indeed, Structural Equation Modelling is a very complex model that has been used to examine complex causal relationships.

5). I am not sure the data on religious beliefs adds anything to the paper.

>>>Thank you. The authors considered the data on religious belief to be important since religion might sometimes affect how an individual might perceive life as well as work-related issues. We do thank again the reviewer for the valuable contribution towards improving this manuscript.

Reviewer 2 Report

Please kindly find my comments in the attached file.

Author Response

Dear reviewer,

Thanks for your valuable comments and the time you took to review our manuscript. We have carefully considered your comments and incorporated them into our revision (See also the revisions in tracked changes). We also note that the manuscript has been checked for grammatical mistakes, spelling, punctuation and improvement of style using a professional English editing services.

We have replied your comments by point as below. Please note the following:

-          Our responses herein follow >>>

Yours sincerely,

Yi-Hsin Elsa Hsu

================

Thank you for allowing me to review the paper titled “Perceived Impact of Taiwan’s National Health Insurance Allocation Strategy: Health Professionals’ Perspective”. This is an important topic and it requires investigation given the scarce resource in the health sector of Taiwan. I really enjoyed reading this paper. However, I have a number of comments.  Kindly find my comments for each section separately below.

 >>>We also thank the reviewer for the important suggestions and contributions towards improvement of the manuscript.

Abstract:

The abstract is not very much informative. First, clearly indicate the challenges associated with the implementation of global budgeting within the NHI of Taiwan. Second, show that there is literature gap. Briefly, introduce the method (data including timeframe, statistical approach). Minimize the use of technical words and numbers when presenting your results in the abstract. Readers of the abstract should get a clear picture of your findings. Finally, show how your results inform policy. The current abstract is very crowded and dominated by words that is unclear for the readers. Especially lines 17-24 is very ambiguous.  Consider rewriting the abstract while keeping the above comments.

 >>>Thank you for your suggestions. We have revised the abstract as suggested. However, concerning the results presented in the abstract (lines 17-24), these are the main relationships between key latent variables (as enclosed in ‘…’). The abstract has therefore been revised as below:

Abstract: Studied on health care demand have indicated high levels of public satisfaction with Taiwan’s National Health Insurance (NHI). Though, the global budget allocation mechanism (GBAM) used by NHI led to various adjustments in the providers’ way of practice, quality of care, utilization of care and health expenditure. Studies focusing on the satisfaction of providers with health care supply however remain limited. We therefore explored provider’s perceive impact of the NHI allocation plan. A cross-sectional data of 299 health professionals was collected at Taipei Medical University Hospitals in April 2012. Perceptions and attitudes were assessed using a validated 5-point Likert-type questionnaire before using a structural equation modeling technique to explore the complex interrelationships of NHI’s perceived impact. The causal path relationships between the latent variables ‘characteristics of NHI’s allocation plan’ and ‘perceived positive effect’ (β = 0.39), ‘perceived positive effect’ and ‘satisfaction of health professionals’ (β = 0.53), and between ‘characteristics of NHI’s allocation plan’ and ‘satisfaction of health professionals’ (β = 0.30) were positively associated; while the path relationships between latent variables ‘perceived negative effect’ and ‘satisfaction of health professionals’ (β = -0.27) and  ‘characteristics of NHI’s allocation plan’ and ‘attitude toward allocation criteria’ (β = -0.22) were negatively associated. These results indicate that providers perceived a positive impact of NHI allocation strategy. The NHI allocation plan is an important decision-making tool among policy makers since it helps to optimize outcomes. A research based on its impact at both horizontal and vertical levels on the supply side may be useful towards understanding Taiwan’s GBAM. Policy-makers should therefore consider understanding the impact of GBAM at both the demand- and supply-side in adjusting allocation criteria.

Introduction:

Line 35-37: Please include the following citations 

 Liu, Y., Hsiao, W. C., & Eggleston, K. (1999). Equity in health and health care: the Chinese experience. Social Science & Medicine, 49(10), 1349-1356.

 White-Means, S., & Osmani, A. (2018). Affordable Care Act and Disparities in Health Services Utilization among Ethnic Minority Breast Cancer Survivors: Evidence from Longitudinal Medical Expenditure Panel Surveys 2008–2015. International journal of environmental research and public health, 15(9), 1860.

 >>>Thank you for the suggested references. The authors have decided to include the references in the manuscript.

Line 42-44: What is the source of increased health expenditure? Is that because of increased utilization of services? Aging population? Leakage in the public expenditure? Lack of Technical or Allocative efficiency? Please carefully respond to each of these concerns and include your response in the text as needed. 

 >>>Thank you. As it is documented, the increased expenditure in Taiwan was as result of “increased utilization.” This has also been clearly added in the manuscript.

Line 54-56: Is that really free access to healthcare services? There is no free “lunch”. Please correct the sentences as someone is paying for the health care services.

 >>>Yes, it is free access to healthcare services. However, the NHI relies on the premiums collected from the population, employers and government and this has been clearly stated in the manuscript.

Line 60-61: Please cite the following papers on health expenditure:

 Okunade, A. A., Karakus, M. C., & Okeke, C. (2004). Determinants of health expenditure growth of the OECD countries: jackknife resampling plan estimates. Health Care Management Science, 7(3), 173-183.

Okunade, A. A., & Osmani, A. (2018). Technology, Productivity, and Costs in Healthcare. In Oxford Research Encyclopedia of Economics and Finance (pp. 1-21). Oxford: Oxford University Press.

Goddard, M., & Smith, P. (2001). Equity of access to health care services: Theory and evidence from the UK. Social science & medicine, 53(9), 1149-1162.

 >>>Thank you for the suggested references. The authors have decided to consider the references in as suggested by the reviewer.

Please replace subtitle 1.1 (line 61) with the main title: Institutional setting and background

>>>Thank you. The authors are of the opinion that the subtitle 1.1 should not be modified, and instead it should be “Taiwan NHI’s risk pooling and allocation strategy.” This is because the main focus of the study was on the resource allocation strategy as has been elaborated in the very sub-section. Otherwise, thanks again for the comment and insight.

under the new title make sure to fully explain the healthcare system in Taiwan and also provide a complete background of the Social Health Insurance system in the country.

 >>>Thank you for your suggestion. The authors briefly discussed the healthcare system in the “Introduction” section. However, Taiwan’s healthcare system has been adequately explained in another publication, as has been indicated and referenced (Wu, T. Y. et al, 2010).

Line 74-77: Why the government share in health expenditure is more than doubled?  Economic growth? Increased prevalence and incidence of diseases?  Aging population? Mis-allocation of resources? Leakage in expenditure? Please provide a thorough justification for this part?

 >>>Thank you. This was as a result of increased utilization of health care services as has been indicated in the manuscript clearly in the very section.

Who is health professional? Make sure to provide a complete definition of health professionals? Doctors, nurses, midwives? technicians? health economists?

>>>Thank you. The complete definition of health professionals has been provided as suggested by the reviewer as physicians, nurses, pharmacists, nutritionists, laboratory technologists and others.

Line 95-108: Please make a table and show the allocation of resource for a year, say 2017 as an example. This will provide more intuitive picture of resource allocation in the NHI.

 >>>Thank you. The formula only gives the indicators that are considered in the global budget and does not give the actual spending/allocation. We therefore considered citing a published paper by Chen and Chuang that has already given the details of recent medical spending. Thanks again for the reviewer’s contributions.

Method:

 Line 124-137: Do you think the result from this analysis is generalizable to entire country? It seems that the data is restricted only one hospital with 3000 beds.

>>>Thank you. The result of this study is not generalizable to entire country, but it only gives a head start into the discussions pertaining to the NHI’s impact from the perspective of the health care supply side. We had also clearly indicated this limitation, generalizability, among our study limitations.

What is the definition of gatekeeper in Taiwan’s health care system? In US, for instance, Primary care Provider (PCP) acts as system gatekeeper. Explicitly explain the meaning of gatekeeper for the Taiwan context.

>>>Thank you. The authors have decided to replace the term gatekeeper with health care providers since the term might confuse the readers depending on the context. Unlike the UK system that has a much stronger focus on the gatekeeper’s role on primary care, the Taiwanese system allows for greater patient choice.

Don’t you think that health professional who are allocating their resource (time) three different hospital would be exhausted and their perceived understanding would be very much biased? Do you think that the result would be very much different if you perform this analysis for a hospital setting where staff are exclusive works at one spot?

>>> Thank you. Indeed, the authors have considered that the reviewer’s comment might be true and have added it among our limitations.

2.2  Structural model and instruments:

What is the base for selection of dependent and independent latent variables? Any citation or reference for that?

>>>Thank you. We based our selection of the dependent and independent latent variables on a previous study that has been cited accordingly as suggested by the reviewer.

Results:

Please explain your results in two subsections

 Descriptive statistics and Structural equation modelling result.

>>>Thank you. We have considered the reviewer’s recommendations and adjusted the sections accordingly.

Also, in the current version of the paper, the result looks like a “laundry list” without a lot of logical consistency and explanations. Please follow the guidelines on how to write an attractive result section.

>>> Thank you. The flow of the result section is important and this has been taken into consideration. However, it is important to note that when reporting SEM results, the fit indices, parameter estimates and structural model estimates are always reported.

Why do you think religious status matters? Explain it in detail.

>>>Thank you. The authors considered the data on religious belief to be important since religion might sometimes affect how an individual might perceive life or even work-related issues. This data was collected and indeed we did find that some participants did not have any religious affiliation.

You provided the estimates both in tables and schematic. Why is that? Is that standard? Which one is more representative and easier to understand? What is meaning of those arrows?

>>>Thank you. Yes, reporting of our results, in both SEM and tables, provides a clear understanding for the general readership who may not be in this particular field and may not understand SEM. We have also added a note in the figure explaining the meaning of the arrows.

Do estimates of SEM provides causal inference or just have correlative values? Please explicitly explain this in the text.

>>>Yes, the SEM provides the causal inference as the reviewer has indicated. We have also clarified this in-text.

Conclusions:

 It is much better not to end the conclusion of the paper with limitations (negative dimensions of the paper), please talk about overall policy implication of the paper in the last paragraph.

>>>Thank you for the suggestion. The reviewer’s suggestion has been by the authors and the limitations have been moved to the discussion section while policy implication has been added in the conclusion section.

Finally, make sure to provide at least 10 citations from Economics Journals given that the paper is submitted into health economics section. There a number of good economic papers on health expenditure, healthcare equity, social health insurance and payment mechanisms.

>>>Thank you. We indeed cited only relevant references that were appropriated for our study most of the citations were targeting health economic-related issues.

Please make sure to provide detailed answer for each of my questions and include your answers in the main-text if needed.

>>>We thank the reviewer for the contribution towards the improvement of this manuscript. Thanks again.

Round 2

Reviewer 2 Report

I do not have any other suggestion at this stage.